# Mechanical and Thermal Properties of Thermally Conductive Enhanced Paraffin/Gypsum Composites

**Chang Chen** [1] , **Huan Wang** [1], **Yubin Wang** [2], **Yanxin Chen** [1,*] **and Shaowu Jiu** [1,*]

1   College of Materials Science and Engineering, Xi'an University of Architecture & Technology, Xi'an 710055, China
2   School of Resources Engineering, Xi'an University of Architecture & Technology, Xi'an 710055, China
*   Correspondence: chen_yanxin@xauat.edu.cn (Y.C.); jiushaowu@xauat.edu.cn (S.J.);
     Tel.: +86-133-8921-2279 (S.J.)

**Abstract:** The low thermal conductivity of gypsum-based composites containing phase change materials (PCMs) has limited their application in construction materials. In this study, the mechanical and thermal properties and microstructures of paraffin (PA)/gypsum composites containing iron powder, copper powder, and expanded graphite were characterized by a universal testing machine, thermogravimetry, differential scanning calorimetry, and scanning electron microscopy. The PA/gypsum composites had optimal mechanical properties when the PA content was 20 wt.%. The compressive and flexural strengths were 9.46 MPa and 3.57 MPa, respectively. When the copper powder content increased, the densities and porosities of the PA/gypsum composites containing the copper powder did not largely change. The average density and porosity of the composites were 1.17 g/cm$^3$ and 46%, respectively. The compressive and flexural strengths of the PA/gypsum composite containing 8 wt.% copper powder were optimal. A phase change temperature of 48.8 °C was obtained when 8 wt.% copper powder was added to the PA/gypsum composite. The thermal conductivities of the PA/gypsum composites were lower than those of PA/gypsum composites with 8 wt.% iron powder, 8 wt.% copper powder, or 8 wt.% expanded graphite. Parts of coarse calcium sulfate dihydrate crystals were present at the interfaces between the gypsum and PA, which indicated that the growth of calcium sulfate dihydrate crystals was affected by the copper powder.

**Keywords:** paraffin/gypsum composites; thermally conductive reinforcing materials; mechanical properties; thermal properties; microstructures

## 1. Introduction

Building energy consumption, which accounts for more than 50% of the total energy consumption in China, is increasing with the continuously increasing requirements for the comfort of building and indoor environments. The high amount of energy consumption in buildings can lead to serious consequences, such as energy depletion and climate change [1]. Therefore, the development and application of energy storage technologies and energy-saving materials are of theoretical significance as an effective approach toward building energy conservation and the development of green buildings [2–5]. Phase change materials (PCMs) can be used to increase the thermal mass of new and retrofit applications. The energy transfer in PCMs occurs through a change in the state of the PCMs from solid to liquid or vice versa [6]. In the phase change process, the phase change material absorbs heat or cold from the surrounding environment and releases the energy required by the environment in the next phase change process, thus controlling the temperature difference of the surrounding environment without fluctuating too much. PCMs have been widely reported as ideal materials for the moderation of indoor air temperatures in residential buildings and shifting of cooling loads to off-peak periods [7,8]. Therefore, many researchers have analyzed different types of PCMs and investigated their use in

the construction of envelopes for this purpose [9–11]. In addition, the decrease in natural resources and increase in energy demand have led to the requirement for optimal energy storage. PCMs absorb heat during daytime and release heat at night [12,13]. A high heat storage density can be achieved in several temperature ranges, and the phase change mechanism can be used effectively to regulate the energy supply and demand. Recently, different PCMs such as paraffin (PA) and fatty acids such as stearic and palmitic acids have been extensively studied [14–17]. Heat energy can be exchanged with the surrounding environment through phase transformation, which can be used to optimize the thermal comfort of buildings [18].

Gypsum is an important construction material because of its low cost, large number of microporous structures, good sound insulation, fireproofing and energy storage properties, and frequent use with a PCM [19]. The use of gypsum-based materials such as adhesives, plasters, block elements, wall panels, ceilings, and floor coverings is increasing in construction [20]. Flue gas desulfurization gypsum has a good heat preservation performance, relatively low firing temperature, relatively high strength, relatively light finished product quality, less environmental pollution, its adaptability is better, and it is easy to combine with other materials to prepare construction materials with excellent performance compared with other construction materials, so its application range is more extensive. In addition, flue gas desulfurization gypsum as a kind of solid waste not only has the same physical composition as natural gypsum, but even has better performance than natural gypsum in some aspects. Therefore, its application in building materials not only realizes the large-scale resource utilization of solid waste, but also solves the large demand for construction wall materials for social development, and at the same time strongly inhibits the over-exploitation of mineral resources and reduces the burden on the environment. Thus, gypsum is an ideal candidate for the development of new green building materials for thermal management applications in building envelopes [21,22]. As an organic PCM with a high latent heat and stable physicochemical properties, PA is widely used as a building filler for the absorption and storage of solar energy [23]. In addition, the internal PA transforms into the liquid phase, migrates through the openings in the gypsum matrix, and continuously adheres to the surface of the $CaSO_4 \cdot 2H_2O$ crystals. The PA forms a three-dimensional network structure, which improves the strength of the composite [19]. However, the thermal conductivity of gypsum is low (0.33 W/(m·K)), which hinders the external heat transfer to the PCMs in time; consequently, the energy storage efficiency is low [24].

Highly thermally conductive materials such as metal powders, graphite, carbon fibers, titanium dioxide, and titanium nitride are used to improve the thermal conductivities of PCMs [25–30]. Expanded graphite (EG) has been widely used for the preparation of shape-stabilized composite PCMs owing to its high adsorption capacity, low density, low cost, and noncomputability [31–33]. It is most often used in combination with organic PCMs, such as PA [34–36], owing to its good thermal conductivity [37,38]. In synthetic composite PCMs, EG contributes to the shape stability and to improving the thermal conductivity owing to the interconnected three-dimensional graphitic thermal conductivity networks [34,39–41]. Mohamed et al. [42] showed that the addition of nano alumina significantly improved the thermal conductivities of PA and microcrystalline wax composites. Numerous experiments have shown that the incorporation of nanoparticles improves the thermal conductivity as well as heating and cooling rates of organic PCMs [43]. Sahan et al. [44] used a dispersion technology to prepare a PA/$Fe_3O_4$ composite PCM. The thermal conductivity of the PCM increased by 48% and 60% when the $Fe_3O_4$ content in the composite was 10% and 20%, respectively. Goli et al. [45] studied the influence of graphene on the thermal conductivities of PA-based PCMs. When the graphene content was 1% and 20%, the thermal conductivity of the graphene–PA composite PCM increased by 15 and 45 W/m·K, respectively. Cheng et al. [46] used graphite, PA, and high-density polyethylene to prepare thermal-conductivity-stereotyped PCMs. The thermal conductivity of a PA/high-density polyethylene composite PCM was increased by 67.74% when the graphite content was 16%.

In this study, PA/gypsum composites were prepared using PA as the PCM and gypsum as the matrix. The optimal PA/gypsum composites were obtained by analyzing the physical and mechanical properties of the composites. Thermal conductors (TCs) such as copper powder, iron powder, and EG were then used as thermally conductive reinforcing materials to improve the thermal conductivity of the optimal PA/gypsum composite. The physical, mechanical, and thermal properties and microstructures of the thermally conductive enhanced PA/gypsum composites were analyzed by the flexural and compression testing machine, differential scanning calorimeter, thermal constant analyzer, and scanning electron microscope.

## 2. Materials and Methods

### 2.1. Raw Materials

The flue gas desulfurization gypsum was purchased from Sunzhen Power Plant in Xi'an. The main component was $CaSO_4 \cdot 2H_2O$ and the density was 2.32 g/cm$^3$. PA was produced by Sinopharm Group Chemical Reagent Co. The main component was a straight-chain alkane with a molecular formula of $C_nH_{2n+2}$, density of 0.90 g/cm$^3$, particle size of 250–500 μm, and melting point of 47–64 °C. The iron powder, copper powder, and EG were produced by Tianjin Tianli Company. Their densities were 7.85, 8.92, and 0.60 g/cm$^3$, respectively; thermal conductivities were 80, 401, and 300 W/m·K, respectively; and particle sizes were approximately 70–100 μm.

### 2.2. Sample Preparation

The water-plaster ratio for the given gypsum material was 0.69, as per the experimental requirements [18]. The gypsum blocks were prepared in accordance with the standard GB/T 17669.3–1999 "Determination of mechanical properties of gypsum". PA and water were added to the gypsum. The content of PA was 5, 10, 15, 20, and 25 wt.%. The specific compositions of the PA/gypsum composites with dimensions of 40 mm × 40 mm × 160 mm are shown in Table 1. Then, the copper powder, iron powder, and EG were mixed with PA/gypsum composites at a certain content relative to the mass of gypsum to produce the TC/PA/gypsum composites. The compositions of the TCs are shown in Table 2. The physical, mechanical, and thermal properties of the composites were then tested, and the optimal amounts of PA and thermally conductive reinforcing materials were determined. The amount of gypsum, PA and TCs and the ratio of water to paste were calculated according to the designed ratio of the experiment. Firstly, the pre-weighed solid powder was put into the barrel and mixed well, and then the mixing water was added and mixed for 120 s and poured into the 300 mm × 300 mm × 300 mm insulation mortar test mold, which was demolded after 2 h. After demolding, it was put into the standard maintenance box at 40 °C for maintenance.

**Table 1.** Specific compositions of the PA/gypsum composites.

| No. | Gypsum (g) | Paraffin (wt.%) | Water (g) |
|:---:|:---:|:---:|:---:|
| 1 | 1000 | 5 | |
| 2 | 1000 | 10 | Normal consistency |
| 3 | 1000 | 15 | water requirement of |
| 4 | 1000 | 20 | gypsum |
| 5 | 1000 | 25 | |

**Table 2.** TC contents in the TC/PA/gypsum composites.

| TC | Paraffin (wt.%) | Content (wt.%) | | | | |
|:---:|:---:|:---:|:---:|:---:|:---:|:---:|
| Iron powder | | 2 | 4 | 6 | 8 | 10 |
| Copper powder | The optimum content determined according to Table 1 | 2 | 4 | 6 | 8 | 10 |
| EG | | 2 | 4 | 6 | 8 | 10 |

*2.3. Material Characterization*

The masses of the PA/gypsum and TC/PA/gypsum composites were evaluated using a scale with an accuracy of 50 g, and then the length, height, and thickness of the specimen were measured according to JC/T 689–2010 "Gypsum blocks". The density $\rho_0$ was calculated according to Equation (1), accurate to 1 g/cm$^3$:

$$\rho_0 = \frac{G}{L \times H \times T},$$ (1)

where $\rho_0$ is the apparent density, $G$ is the specimen mass, $L$ is the specimen length, $H$ is the specimen height, and $T$ is the specimen thickness.

The porosities of the PA/gypsum and TC/PA/gypsum composites were determined by Equation (2):

$$P = \frac{V_0 - V}{V_0} = 1 - \frac{V}{V_0} = 1 - \frac{\rho_0}{\rho},$$ (2)

where $P$ is the porosity, $V_0$ is the volume in the natural state, $V$ is the volume in the absolute compact state, $\rho$ is the calculated density of PA/gypsum or TC/PA/gypsum.

The mechanical properties of the two types of gypsum composites were investigated using a flexural testing machine (KZJ–5000, Shenyang Great Wall Electromechanical Equipment Factory, Shenyang, China) and compression testing machine (JYE–2000, Wuxi Building Materials Instrument Factory, Wuxi, China). Ten samples were prepared for each size of each type. The performances of the composites prepared with different contents of PA and TC were investigated.

The thermal properties of the TC/PA/gypsum composites were measured using differential scanning calorimetry (DTG–60H, Shimadzu Company; DSC, Netzsch 200F3) in a nitrogen atmosphere at a rate of ±10 °C/min in the range 10–120 °C. A thermal constant analyzer (TPS2500S, Hot Disk Co., Ltd., Göteborg, Sweden) was used to evaluate the average thermal conductivities of the PA/gypsum and TC/PA/gypsum composites; three samples were prepared for each size of each type of the composites. In accordance with the standard GB/T 10294-2008 "Determination of steady-state thermal resistance and related characteristics of insulation materials Protective hot plate method", the protective hot plate method is used to test the thermal conductivity of insulation materials based on the principle of steady-state heat transfer for TPS2500S building materials thermal conductivity testing equipment. One side of the specimen is the hot plate and the other side is the cold plate, and a uniform plate specimen with a certain thickness and parallel surfaces is sandwiched between the two plates. The one-dimensional constant heat flow of the infinite plate bounded by two parallel uniform temperature plates in the ideal state is then established by setting the temperature of the two plates, and the heat flow rate $Q$ is obtained by measuring the heat flow rate of the central metering plate of the hot plate after reaching the steady state; finally, the thermal conductivity of the specimen is obtained according to the calculation formula of thermal resistance $\lambda$. The TC/PA/gypsum composites were analyzed by scanning electron microscopy (SEM, Quant200, FEI Company, Hillsboro, OR, USA). The acceleration voltage of the SEM was 25.00 kV, and the samples were coated with conductive gold before performing the measurement.

## 3. Results and Discussion

*3.1. Physical and Mechanical Properties of PA/Gypsum Composites*

The densities, porosities, 7-day (d) compressive strengths, and 7 d flexural strengths of the gypsum-based composites with different PA contents are shown in Figure 1. The density of the PA/gypsum composites did not change largely in the PA content range of 5 to 20 wt.%. The composites had the lowest density of 1.11 g/cm$^3$ when the PA content was 25%. With the increase in the PA content, the porosities of the PA/gypsum composites first decreased and then increased. The paraffin was dense and the particle size (250–500 μm) was much larger than the diameter of micropores in the gypsum matrix

(5–10 μm). When the paraffin content increased, the amount of micropores per unit volume of paraffin/gypsum composites reduced, and therefore the porosity of the paraffin/gypsum composites decreased. However, when the paraffin content exceeded 20%, some of the spherical paraffins were superimposed and new pores were formed, which leads to an increase in porosity. So, when the PA content was 20 wt.%, the porosity of the composite had the lowest value of 45%. The densities of the PA/gypsum composites should theoretically gradually decrease with the increase in the PA content because of the low density of PA. However, the lowest porosity was obtained when the PA content was 20 wt.% because some PA particles were present in the macropores of the gypsum.

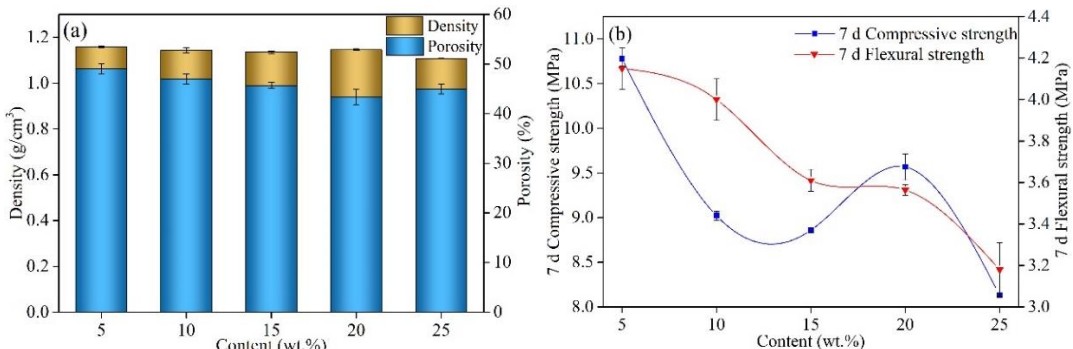

**Figure 1.** (**a**) Densities, porosities, (**b**) 7 d compressive strengths, and 7 d flexural strengths of the PA/gypsum composites.

The 7 d compressive strength of the PA/gypsum composites decreased, increased, and then decreased again as the PA content increased from 5 to 25 wt.%, as shown in Figure 1b. The 7 d flexural strength of the composites decreased with the increase in the PA content. However, the flexural strengths of the composites with PA contents between 15 and 20 wt.% were not largely different. Therefore, considering that the composites have certain heat storage and exothermic capacities, the PA/gypsum composites had the optimal mechanical properties when the PA content was 20 wt.%. The compressive and flexural strengths were 9.46 and 3.57 MPa, respectively. Zhang [47] prepared a microencapsulated phase change material with silica as the shell layer and PA as the core material, and then compounded the microcapsules into gypsum. The gypsum material had high mechanical strength, denseness, and mechanical properties when the microcapsules content was 10 wt.%. However, the optimum content of paraffin in this study was 20 wt.%, which made the PA/gypsum composite have better thermal storage properties in comparison.

### 3.2. Physical and Mechanical Properties of Thermally Conductive Enhanced PA/Gypsum Composites

The densities, porosities, 7 d compressive strengths, and 7 d flexural strengths of the TC/PA/gypsum composites are shown in Figure 2. As shown in Figure 2a, the densities and porosities of the PA/gypsum composites containing iron powder slightly changed as the iron powder content increased. The average density and porosity of the composites were 1.14 g/cm$^3$ and 45%, respectively. The 7 d compressive strength of the PA/gypsum composites containing iron powder decreased, increased, and then decreased again. The 7 d flexural strength of the composites had a similar irregular trend, as shown in Figure 2b. The changes in the maximum and minimum values of the 7 d compressive and flexural strengths of the PA/gypsum composites containing iron powder were not significant.

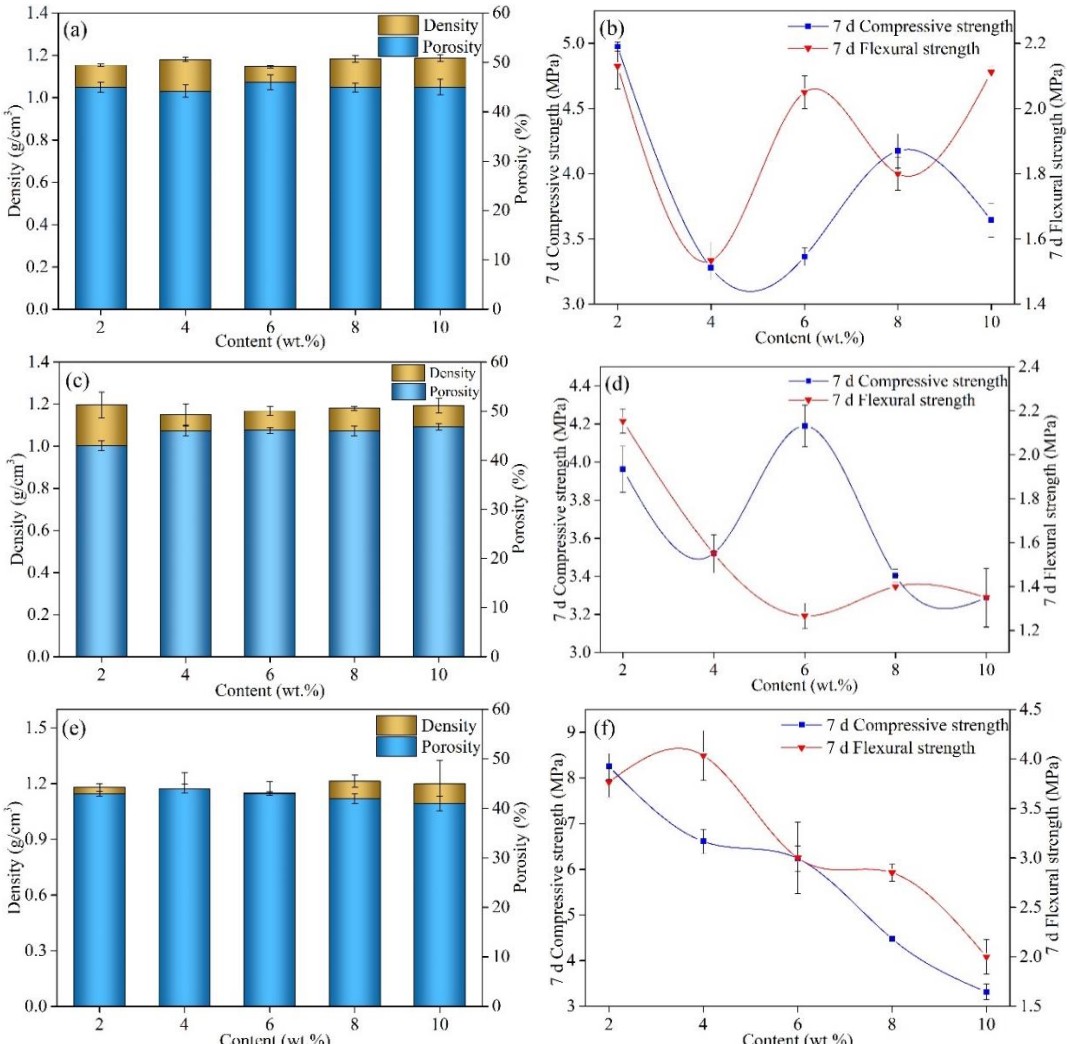

**Figure 2.** Densities, porosities, 7 d compressive strengths, and 7 d flexural strengths of PA/gypsum composites mixed with (**a**,**b**) iron powder, (**c**,**d**) copper powder, and (**e**,**f**) EG.

As shown in Figure 2c,d, when the copper powder content increased, the densities and porosities of the PA/gypsum composites containing copper powder did not change considerably. The average density and porosity of the composites were 1.17 g/cm$^3$ and 46%, respectively. The trends of the mechanical properties of the PA/gypsum composites containing copper powder were similar to those of the PA/gypsum composites containing iron powder. With the increase in the copper powder content, the 7 d compressive strength of the PA/gypsum composites containing copper powder decreased, increased, and then decreased again, while the 7 d flexural strength of the composites decreased. The changes in the maximum and minimum values of the 7 d compressive and flexural strengths for the PA/gypsum composites containing copper powder were also not significant. The metal powders resulted in a poor interface bonding of the TC/PA/gypsum composites, which weakened the mechanical properties of the composites.

The densities of the PA/gypsum composites containing EG increased with increasing EG content. The porosities of the composites changed slightly, as shown in Figure 2e. The maximum density of the composites was 1.21 g/cm$^3$, while the average porosity was 41%. The 7 d compressive and flexural strengths generally decreased with the increase in the EG content. When the EG content was 10 wt.%, the 7 d compressive and flexural strengths of the composites were 3.32 and 2.07 MPa, decreased by 59.8% and 45.1%, respectively, compared to the composites with 2 wt.% EG. Because the EG had a poor bonding ability with

PA/gypsum composites due to its rich pore structure, softness, resilience, self-adhesiveness, and low density, the mechanical properties of the PA/gypsum composites containing the EG were weakened. However, the 7 d flexural strengths of the PA/gypsum composites containing 6 and 8 wt.% EG were did not vary significantly. Therefore, the optimal PA/gypsum composite containing 8 wt.% TC was used in the subsequent experiments.

### 3.3. Microstructures of the Thermally Conductive Enhanced PA/Gypsum Composites

A thermal analysis of the PA/gypsum and TC/PA/gypsum composites was conducted. The thermogravimetry and DSC curves are shown in Figure 3. The mass losses of the 20 wt.% PA/gypsum composite, PA/gypsum composite with 8 wt.% iron powder, PA/gypsum composite with 8 wt.% copper powder, and PA/gypsum composite with 8 wt.% EG were 0.76%, 0.91%, 0.93%, and 1.60%, respectively. The mass losses of the three types of TC/PA/gypsum composite were larger than that of the PA/gypsum composites because the metals and EG were high-thermal-conductivity materials, and thus heat was easily transferred from the surface to the interior of the composites.

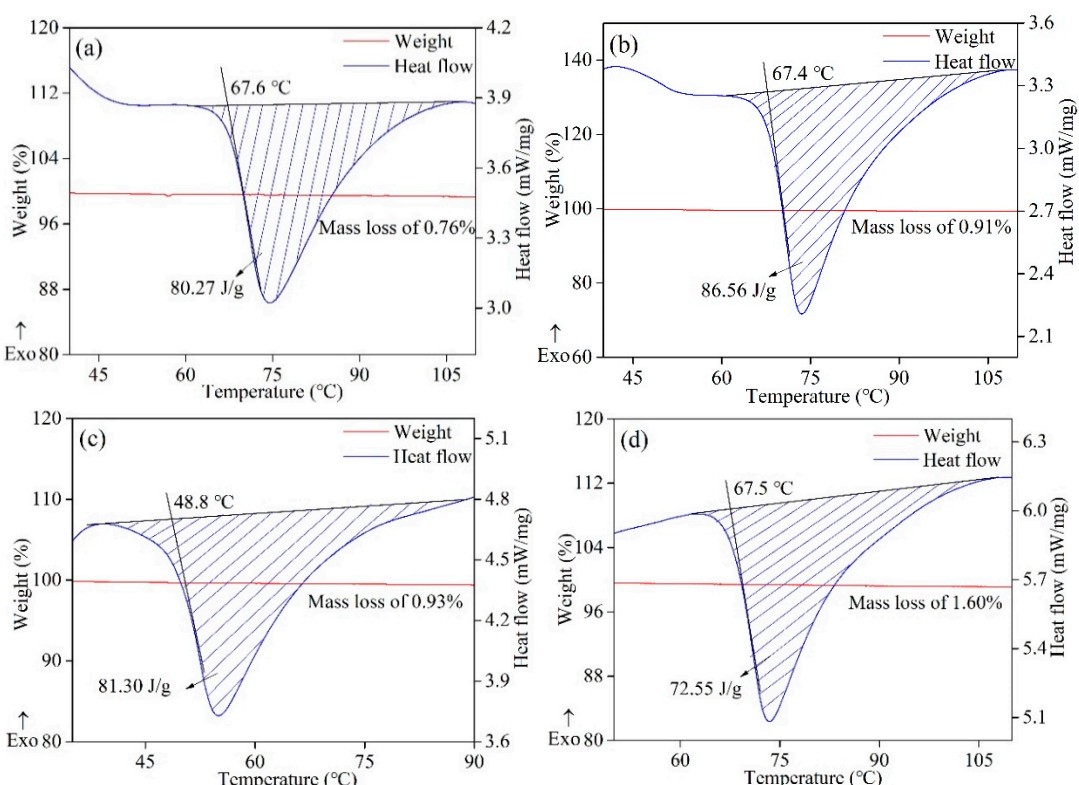

**Figure 3.** Thermal analysis curves of the PA/gypsum and TC/PA/gypsum composites. (**a**) PA/gypsum composites; (**b**) PA/gypsum composites with 8 wt.% iron powder; (**c**) PA/gypsum composites with 8 wt.% copper powder; and (**d**) PA/gypsum composites with 8 wt.% EG.

As shown in Figure 3a,b,d, the phase change temperatures were approximately 67 °C for the PA/gypsum composite, PA/gypsum composite with 8 wt.% iron powder, and PA/gypsum composite with 8 wt.% EG, respectively. However, a phase change temperature of 48.8 °C was obtained when the 8 wt.% copper powder was added to the PA/gypsum composite (Figure 3c). The phase change enthalpies of the PA/gypsum composite, PA/gypsum composite with 8 wt.% iron powder, and PA/gypsum composite with 8 wt.% copper powder were 80.27, 86.56, and 81.30 J/g, respectively. For the PA/gypsum composite with 8 wt.% EG, the phase change enthalpy of 72.55 J/g was lower than those of the other composites because of the low density of EG, which led to a smaller effective amount of PA per unit volume. Therefore, the thermal properties of the PA/gypsum

composite with 8 wt.% copper powder were optimal according to its low mass loss, low phase change temperature, and high phase change enthalpy.

Figure 4 shows the thermal conductivities of the PA/gypsum and TC/PA/gypsum composites. The thermal conductivity of the PA/gypsum composites was lower than those of the PA/gypsum composites with 8 wt.% iron powder, 8 wt.% copper powder, and 8 wt.% EG. The thermal conductivity of the copper powder (401 W/m·K) was higher than that of the iron powder (80 W/m·K). Thus, the thermal conductivity of the PA/gypsum composites with 8 wt.% copper powder was 14.6% higher than that of the PA/gypsum composites with 8 wt.% iron powder, as the copper powder had a high thermal conductivity and similar density (iron powder: 7.85 g/cm$^3$; copper powder: 8.92 g/cm$^3$). The PA/gypsum composite with 2 wt.% EG also had a relatively high thermal conductivity of 0.1653 W/m·K because of the thermal conductivity of EG (300 W/m·K) and relatively small mass of PA per unit volume. Therefore, the heat transfer performance of the PA/gypsum composite with 8 wt.% copper powder was optimal. In addition, Khodadadi [27] showed that carbon-based nanostructures and carbon nanotubes exhibited higher thermal conductivity compared to metal/metal oxide nanoparticles. However, nanomaterials such as carbon-based nanostructures and carbon nanotubes are difficult to apply in large quantities due to their high cost.

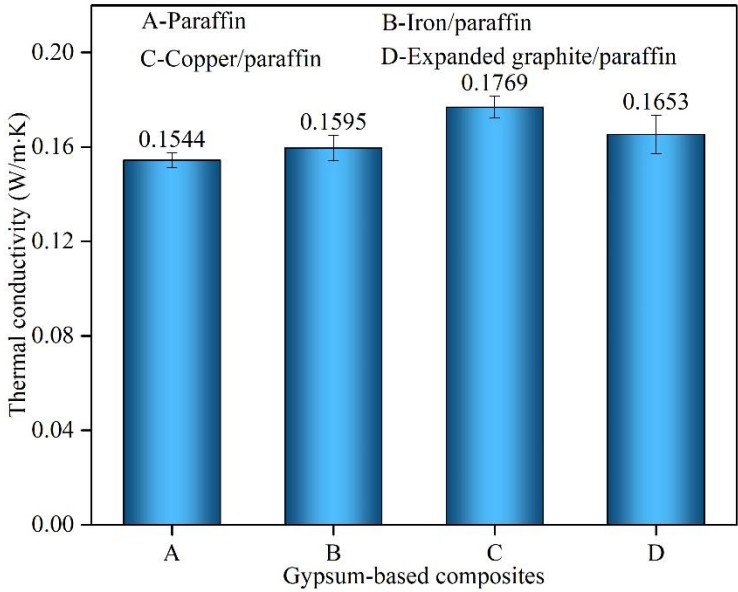

**Figure 4.** Thermal conductivities of the PA/gypsum and TC/PA/gypsum composites.

Fracture morphologies of the cross sections of the PA/gypsum and TC/PA/gypsum composites are shown in Figure 5. The spherical white pits and black parts were left by the PA (Figure 5a). The fracture morphology between the PA and gypsum matrix indicates that the interface was relatively weak. The PA wax with an average particle size of 200 μm was uniformly distributed in the gypsum matrix. A large number of micropores were evenly distributed in the gypsum composite with 20 wt.% PA, which also indirectly indicates that gypsum was porous and could be used as a thermal insulation material. In addition, the short rod-shaped crystals of hydration crystallization products were distributed in a disordered manner according to the enlarged image in Figure 5a.

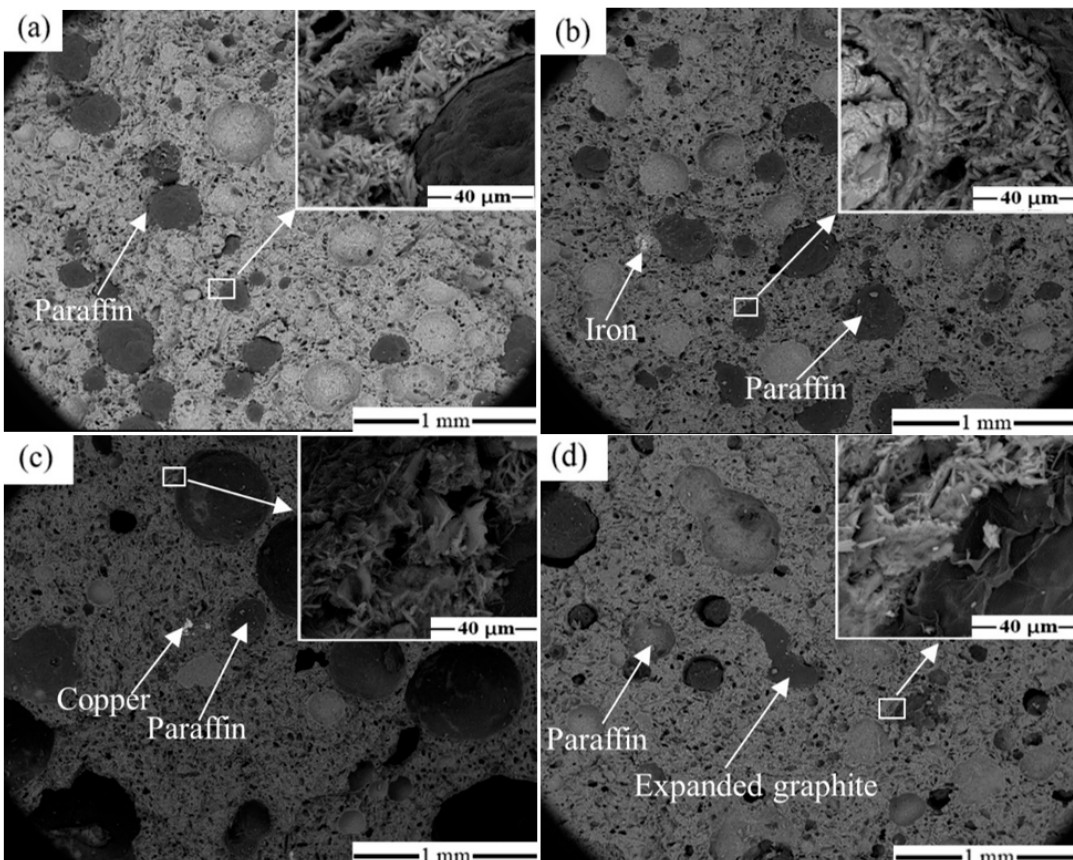

**Figure 5.** SEM images of the PA/gypsum and TC/PA/gypsum composites. (**a**) PA/gypsum composites; (**b**) PA/gypsum composite with 8 wt.% iron powder; (**c**) PA/gypsum composite with 8 wt.% copper powder; (**d**) PA/gypsum composite with 8 wt.% EG.

For the PA/gypsum composites with 8 wt.% iron powder and copper powder, holes could be observed in the composites where the gypsum and PA were combined, and the gypsum crystals were distributed along the direction of PA particles. However, parts of coarse calcium sulfate dihydrate crystals were present at the interfaces between the gypsum and PA, which indicates that the growth of calcium sulfate dihydrate crystals was affected by the copper powder (Figure 5c). The crystal morphologies of the hydration crystallization products for the interfaces of the PA/gypsum and PA/gypsum composites containing 8 wt.% iron powder were similar. Furthermore, the metals had high heat absorption coefficients during heat transfer and brought the heat to the PA and gypsum, thus accelerating the heat transfer and improving the efficiency of the composites.

In Figure 5d, there are some agglomerations of PA because of the flake structure and low density of EG. Additionally, the flocculated hydration crystallization products were present at the interfaces between PA and gypsum, which weakened the mechanical properties of the PA/gypsum composites containing EG. Although the volume content of EG was high compared to those of the iron and copper powders and its thermal conductivity was relatively high, the addition of the EG led to the nonuniform distribution of PA, and thus the thermal conductivity of the composite was lower than that of the PA/gypsum composite with the copper powder.

## 4. Conclusions

The effect of thermal conductors on the thermal properties of paraffin/gypsum composites was investigated in order to increase the thermal conductivity caused by the direct incorporation of organic phase change materials in gypsum, and the variation in the physical and mechanical properties of the composites after paraffin and thermal conductors

migration were investigated. The 7 d flexural strength of the composites decreased with the increase in the paraffin content. However, the flexural strengths of the composites with paraffin contents between 15 and 20 wt.% exhibited negligible change. Therefore, the paraffin/gypsum composites had the optimal mechanical properties when the paraffin content was 20 wt.%. The changes in the maximum and minimum values of the 7 d compressive and flexural strengths for the paraffin/gypsum composites containing copper powder were not significant relative to the changes in the gypsum composite containing 20 wt.% paraffin. The phase change enthalpy of the paraffin/gypsum composite with 8 wt.% expanded graphite was lower than that of the other composites because of the low density of expanded graphite, which led to a smaller effective amount of paraffin per unit volume. The high-performance thermal conductors had good thermal adsorption properties during heat transfer, and they provided heat to the paraffin and gypsum, thus accelerating the heat transfer and improving the efficiency of the composites.

**Author Contributions:** Conceptualization, C.C.; methodology, C.C.; validation, H.W.; formal analysis, C.C. and H.W.; investigation, H.W.; resources, C.C.; data curation, C.C.; writing—original draft preparation, H.W.; writing—review and editing, C.C., Y.W., Y.C. and S.J.; supervision, C.C. and Y.C.; project administration, Y.W. and C.C.; funding acquisition, Y.W., Y.C. and S.J. All authors have read and agreed to the published version of the manuscript.

**Funding:** This research was supported by the National Natural Science Foundation of China (No. 51974218), the Anhui Provincial Science and Technology Major Project (No. 2021e03020003) in China, and the Shaanxi Provincial Innovation Capability Support Program (No. 2021TD-53).

**Institutional Review Board Statement:** Not applicable.

**Informed Consent Statement:** Not applicable.

**Data Availability Statement:** The data used to support the findings of this study are available from the corresponding author upon request.

**Conflicts of Interest:** The authors declare no conflict of interest.

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
