# Peer review of "Mechanical and Thermal Properties of Thermally Conductive Enhanced Paraffin/Gypsum Composites"

_processes, doi:10.3390/pr11040999_

Round 1

Reviewer 1 Report

Dear authors,

you have nicely described the mechanical and thermal properties of thermally conductive enhanced paraffin/gypsum composites.

I have some minor comments in regards to your manuscript:

1. Thermal analysis and SEM equipments were not mentioned in the Materials and Methods section. Please add a small description of the equipment (also mention the Voltage so that a competent researcher can reproduce your SEM images/experiment).

2. Can you give a reason why in, for example Fig 1, with the increase in the PA content, the porosities of the PA/gypsum composites first decreased and then increased?. One would expect that the pores get filled with PA so that the porosities will always decrease. Please just give a reason for this behavior.

3. Please rephrase your conclusions so that it does not look like a laboratory report itemized by 1, 2, 3...

Reviewer 2 Report

Interesting article with practical conclusions. The composition of materials in the tested range of parameters was optimized.

The lack of monotonicity in the case of several dependencies may indicate that the phenomena accompanying the formation of the material are complicated with so many parameters, determined in a random way. In the future, synergy between the composition and the conditions of material formation should be sought.

It is worth noting that materials with the addition of iron, bismuth or tungsten compounds could also replace lead in radiological protection, e.g. in nuclear medicine facilities or dental offices equipped with X-ray machines.

Reviewer 3 Report

The study is about creating conposites using PA and gypsum. Physical and dynamic charactetistics were shown to desctibe the thermal properties.

The study is well described, but in the Results and Discussion section there was no comparison of the results with other studies. It should be added to understand the superiority properties of the produced composites.

Round 2

Reviewer 3 Report

The current version can be acceptable.